# Simultaneous Manipulation of the Temporal and Spatial Behaviors of Nanosecond Laser Based on Hybrid Q-Switching

**Haoxi Yang [1], Yuanji Li [1,2,\*], Wenrong Wang [1], Jinxia Feng [1,2] and Kuanshou Zhang [1,2,\*]**

[1] State Key Laboratory of Quantum Optics and Quantum Optics Devices, Institute of Opto-Electronics, Shanxi University, Taiyuan 030006, China

[2] Collaborative Innovation Center of Extreme Optics, Shanxi University, Taiyuan 030006, China

\* Correspondence: liyuanji@sxu.edu.cn (Y.L.); kuanshou@sxu.edu.cn (K.Z.)

**Abstract:** A hybrid Q-switching method based on a special-shaped saturated absorber was proposed for simultaneous manipulation of the temporal and spatial behaviors of a solid-state pulse laser. The temporal–spatial rate equation model of the laser was given and used to optimize the design parameter of the saturated absorber. Best spatial intensity homogenization performance can be expected using an active-passive hybrid Q-switched laser, comprising a Pockels cell and a cylinder Cr:YAG crystal with one end cut as a spherical concave surface. The optimized laser pulse width could be narrowed to 2.39 ns and the laser radial intensity distribution became quasi-super-Gaussian distribution with a radial intensity distribution ratio of 0.91, while that for the Gaussian beam was 0.84. In principle, the laser coherence can be maintained, and the laser spatial intensity distribution can be kept in a long propagation distance.

**Keywords:** hybrid Q-switching; deformed cylinder saturated absorber; laser intensity distribution manipulation; laser pulse width reduction





## 1. Introduction

Solid-state pulse lasers with narrow pulse width (PW) and uniform intensity distribution have found a great many applications in a wide range of fields, including laser power amplification, laser material processing, imaging, high efficiency fiber injection, and lithography. In particular, for the efficient generation of ultra-high energy lasers in master oscillator power amplifier (MOPA) configuration [1–7], which can be further applied in deep ultraviolet laser generation, long distance laser ranging and laser communication, etc. [8–10], high quality seed lasers, with the properties of uniform intensity distribution and good coherence, are strongly needed to achieve efficient utilization of pump energy inside the side-pumped amplifier modules and coherent amplification. For the temporal shortening of the laser PW, there were many mature techniques, for example the short-cavity lasers employing fast electro-optical (EO) Q-switches or deflectors [11–13], the miniature passive Q-switched lasers based on saturated absorbers [14–16], and the mode-locking lasers [17–19]. However, for the spatial homogenization of the laser energy, for instance getting the top-hat intensity distribution and super-Gaussian distribution, many practical difficulties exist [20]. The external cavity optical shaping systems based on aspheric lens combination [21], microlens array [22,23], and diffractive optical elements [24–26] alter the laser intensity distribution by regulating the propagation direction of each part of the laser. Hence, the laser spatial coherence was usually weakened, or even destroyed in principle; the top-hat intensity distribution or super-Gaussian distribution can only be maintained in a centimeter level distance, and an excess power loss was inevitable. Besides that, the intracavity beam shaping method was also proposed, mainly depending on a periodically swinging cavity mirror. Similarly, the laser was operating in multi-mode with poor coherence and stability.

The dual loss modulation method simultaneously employing two Q-switches, usually an active Q-switch and a saturated absorber, is another popular method for laser PW reduction [27–31]. The temporal behaviors of the laser pulse, both PW and the pulse shape symmetry, can be improved comparing with the EO Q-switched laser for the reason that the rising and falling edges of laser pulse are reduced by saturation absorption effect. In comparison with the passive Q-switched laser with a single saturated absorber, the laser energy fluctuation and time jitter are also optimized since the buildup of laser pulse is dominated by the active Q-switching.

Up to now, an effective method enabling simultaneous manipulation of the temporal and spatial behaviors of nanosecond laser, without the expense of complex structure and weak coherence, is still lacking. There is also no previous research relative to the effect of spatial saturation absorption distribution on the laser transverse mode behavior. In this paper, we propose a hybrid Q-switching method based on the deformed saturated absorber for simultaneous manipulation of the temporal and spatial behaviors of a solid-state pulse laser. The temporal–spatial theoretical model of the laser was given and the optimized design of the laser protocol was demonstrated.

## 2. Temporal–Spatial Rate Equation Model of Hybrid Q-Switched Laser

Figure 1 shows a schematic diagram of the model for a 1.06 μm hybrid Q-switched laser. Pump light with a single photon energy of $h\nu_p$ was coupled into the gain medium with an average radius of $\omega_p$. The gain medium was an $a$-cut Nd:YVO$_4$ crystal with dimensions of $2R_g \times 2R_g \times l$. The resonator consisted of five mirrors, an electro-optic (EO) Q-switch and a Cr:YAG saturated absorber. M$_1$, M$_2$ and M$_3$ were all high reflection coated at 1064 nm, while M$_1$ and M$_2$ were plane mirrors, and M$_3$ was a convex mirror with a curvature radius of Roc1. The output coupler (OC) with a curvature radius of Roc2 had an output coupling reflection of $R$ at 1064 nm. TFP was a 45° thin film polarizer (TFP) enabling $p$-polarization light high transmission and $s$-polarization light high reflection passing. A quarter-wave plate (QWP) and an RbTiOPO$_4$ (RTP) Pockels cell with a quarter-wave voltage of $V_{\lambda/4}$ played the role of EO Q-switch, which was driven by a high-voltage driver with a high-level voltage of $V_{hl}$, high-level voltage duration of $t_{qs}$, and a rise time of $t_{qr}$. SA was a deformed cylinder Cr:YAG crystal; the cylinder part had a thickness of $l_{sa,0}$ and a radius of $R_{sa}$, whilst the concave end had a curvature radius of $RS_{sa}$. L$_1$ was a match lens with the same refractive index with Cr:YAG, while the curvature radius of the convex surface was also $RS_{sa}$. The use of match lens not only simplified the resonator design, but also compensated the space-dependent diffraction introduced by the deformed SA. The whole effective cavity length was $L_{c,eff}$, leading to laser mode radii of $\omega_g$ and $\omega_{sa}$ at gain medium and SA, respectively.

To study the temporal and spatial behaviors of the laser dynamically, the photon number density evolution should be evaluated in different spatial areas. Here, we defined a mesh criterion: on the cross-section of each crystal, a radial mesh was defined consisting of $m$ rings and $q$ angular segments, since both the laser and the crystals were axisymmetric, where $m$ and $q$ were positive integers. As shown in Figure 2, in the sector with a included angle of $\theta = 2\pi/q$ and a radial variation range of 0~$R_g$ ($R_{sa}$), the gain medium was gridded along the radial direction with a step size of $\omega_g/m$; similarly, the saturated absorber was also gridded with a step size of $\omega_{sa}/m$. Based on the fact that the intracavity laser had good spatial coherence, an approximation was made that the evolution of the stimulated emission photons generated in $j$th grid on the gain medium, i.e., in the radial variation range of $(j-1)\omega_g/m$~$j\omega_g/m$, was only related to the loss provided by the $j$th grid on the saturated absorber.

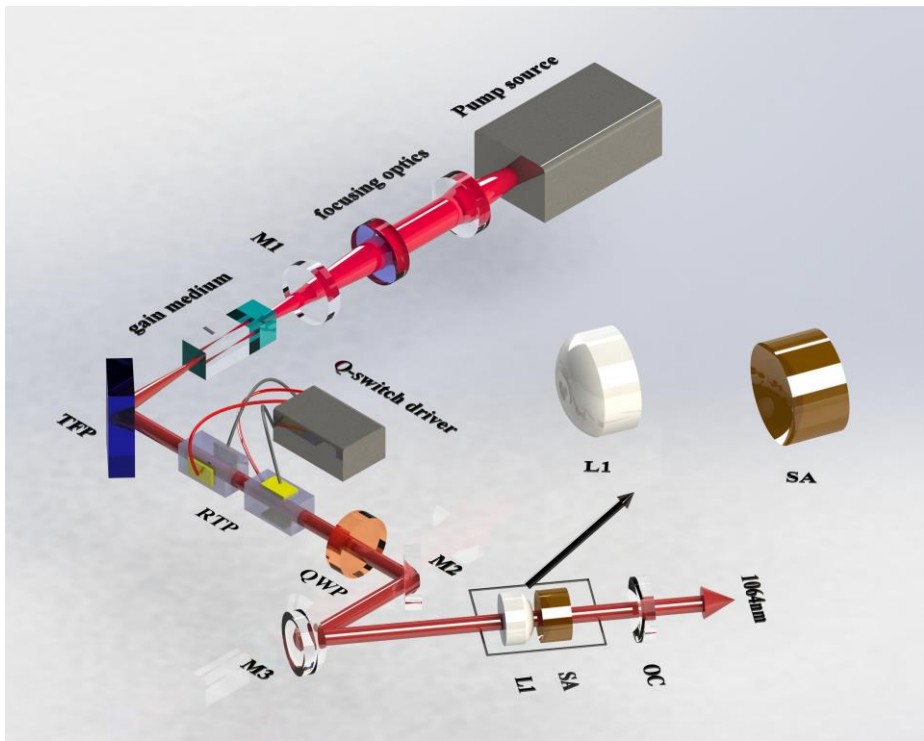

**Figure 1.** Schematic of the 1.06 μm hybrid Q-switched laser. (TFP: thin film polarizer, RTP: Rubidium titanate phosphate crystal, QWP: quarter-wave plate, SA: saturated absorber, OC: output coupler).

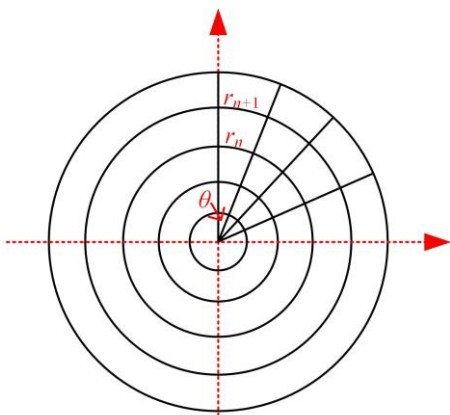

**Figure 2.** Mesh grids of the gain medium and saturated absorber.

For the saturated absorber with the shape shown in Figure 3a, the length of the saturated absorber in the *j*th grid ($l_{sa,j}$) can be approximated to be a constant and represented as:

$$l_{sa,j} = l_{sa,0} + RS_{sa} - \sqrt{RS_{sa}^2 - (j\omega_{sa}/m)^2}. \tag{1}$$

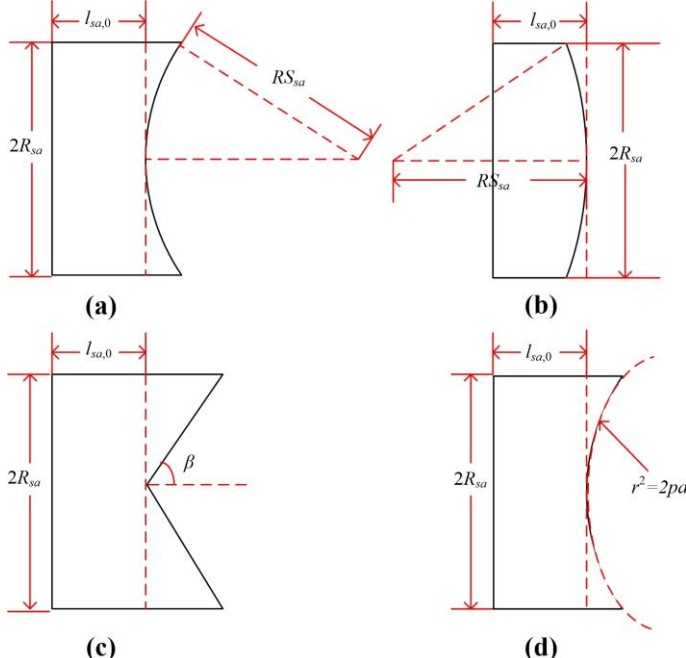

**Figure 3.** Four kinds of shapes of the saturated absorber. (**a**) the cylinder saturated absorber with one end cut as a spherical concave surface; (**b**) the cylinder saturated absorber with one end cut as a spherical convex surface; (**c**) the cylinder saturated absorber with one end cut as a conical surface; (**d**) the cylinder saturated absorber with one end cut as a parabolic surface.

When the shape of saturated absorber was changed to be a deformed cylinder with one end being a spherical convex surface, as shown in Figure 3b, the function expression came to be:

$$l_{sa,j} = l_{sa,0} - RS_{sa} + \sqrt{RS_{sa}^2 - (j\omega_{sa}/m)^2}, \tag{2}$$

For the cases that deformed end of cylinder were cut as a conical surface with an apex angle of $2\beta$, as shown in Figure 3c, and a paraboloid corresponding to a parabolic equation of $r^2 = 2pd$, as shown in Figure 3d, the function expression respectively became:

$$l_{sa,j} = l_{sa,0} + (j\omega_{sa}/m)/\tan(\beta), \tag{3}$$

$$l_{sa,j} = l_{sa,0} + (j\omega_{sa}/m)^2/(2p), \tag{4}$$

It is worth noting that in the following simulations associated with the latter three kinds of SA, the shape of the match lens $L_1$ should be changed to the complementation one. The rate equations of the hybrid Q-switched laser in the $j$th grid can be written as:

$$d\phi_j/dt = 1/t_r\left[2N_j\sigma_{se}l\phi_{g,j} - 2N_{g,j}\sigma_g l_{sa,j}\phi_{s,j} - 2N_{e,j}\sigma_e l_{sa,j}\phi_{s,j} - \left[\ln(1/R) + \delta_0 + \delta_{QS}\right]\phi_j\right], \tag{5}$$

$$dN_j/dt = -N_j c\sigma_{se}\phi_{g,j} - N_j/\tau_f + P_p\eta_p/(\pi\omega_p^2 lh\nu_p), \tag{6}$$

$$dN_{g,j}/dt = -N_{g,j}\sigma_g c\phi_{s,j} + \left(N_{s0,j} - N_{g,j}\right)/\tau_s, \tag{7}$$

$$N_{g,j} + N_{e,j} = N_{s0,j}, \tag{8}$$

where $N_j$ is the population inversion density in the $j$th grid of the gain medium. $N_{g,j}$, $N_{e,j}$ and $N_{s0,j}$ are the ground state ion density, excited state ion density and the total ion density in the $j$th grid of the saturated absorber, respectively. $\phi_j$, $\phi_{g,j}$ and $\phi_{s,j}$ are the average intracavity laser photon density corresponding to the $j$th grid of the gain medium, the laser photon density at the $j$th grid of the gain medium, and the laser photon density at the $j$th grid of the saturated absorber. $t_r = 2L_{c,eff}/c$ is the roundtrip time of the intracavity laser photon. $c$ is the light speed, $\sigma_{se}$ is the stimulated emission cross-section of the gain medium,

$\sigma_g$ and $\sigma_e$ are the ground state and excited state absorption cross-section of the saturated absorber. $\delta_0$ is the intrinsic loss, $\delta_{QS}$ is the time dependent loss of the EO Q-switch. $\tau_f$ is the fluorescence lifetime of Nd:YVO$_4$ crystal, $P_p$ is the incident pump power, $\eta_p = 1 - \exp(-\alpha l)$ is the pump laser absorption efficiency.

Assuming that the pump laser fits top-hat distribution and the intracavity laser fits Gaussian distribution, the initial conditions of the rate equations can be expressed as:

$$N_{0,j} = \frac{P_p T_p \eta_p \eta_f}{h v_p} \frac{\omega_g^2}{\omega_p^2} \frac{1}{\pi \omega_p^2 l}, \tag{9}$$

$$\phi_{0,j} = N_{0,j} \frac{l}{c \tau_f} \frac{d\Omega}{4\pi}, \tag{10}$$

$$N_{g0,j} = N_{s0,j}, \tag{11}$$

where $d\Omega$ represents the solid angle of spontaneous emission that makes contribution to the stimulated emission, $T_0$ is the initial transmissivity of the cylinder saturated absorber corresponding to a thickness of $l_{sa,0}$. $T_p$ is the pump pulse width. $\Delta t$ is the time delay between the rising edge of EO Q switching and the end of the falling edge of the pump pulse.

Moreover, $\delta_{QS}$ can be expressed by the equivalent transmission at the polarizer as:

$$\delta_{QS}(t) = \cos^2\left( \frac{\pi}{2} \frac{V_{hl}}{V_{\lambda/4}} \left(1 - e^{-\left(\frac{t}{t_{qr}}\right)^4}\right)^4 e^{-\left(\frac{t-t_{qs}/2}{t_{qs}}\right)^{400}} \right). \tag{12}$$

### 3. Simulation Results and Discussion

According to Refs [32–35], the function relations between $\sigma_{se}$ and the doped concentration of the Nd:YVO$_4$ crystal ($D_c$), that between $\tau_f$ and $D_c$, as well as that between $\alpha$ and $D_c$ can be expressed as:

$$\sigma_{se} = \left(8.36075 + 2221.49 * D_c - 102818.3 * D_c^2 - 1490820 * D_c^3\right) * 10^{-23}, \tag{13}$$

$$\tau_f = (-1650 * D_c + 106.67) * 10^{-6}, \tag{14}$$

$$\alpha = (13.5 * D_c + 0.8) * 100, \tag{15}$$

Using Equations (1) and (5)–(15), as well as the parameter values given in Table 1, laser evolution in each grid can be numerically simulated. Figure 4a shows the photon density of the emitted pulse laser as a function of radial coordinates, i.e., the lower bound of the grids, at the case of $V_{hl}/V_{\lambda/4} = 1.4$. It can be seen that the local pulse width along the center of the cross-section of the laser is 2.76 ns. Besides that, there exists a falling edge as long as 45 ns for the photons located on the wings of the cross-section. When an aperture is used to block the photons on the wings, almost 61.5% power loss will be introduced. Figure 4b shows the results calculated using $V_{hl}/V_{\lambda/4} = 1$. One can find that better spatial intensity homogenization is achieved. The local pulse width around the central part is narrowed to be 2.39 ns, quasi-super-Gaussian intensity distribution can be realized with the expense of 37% power loss.

**Table 1.** Parameter values used in the theoretical simulations.

| Parameter | Value | Parameter | Value |
|---|---|---|---|
| $l$ | 3 mm | $\omega_g$ | 230 μm |
| $L_{c,eff}$ | 120 mm | $\omega_{sa}$ | 2.5 mm |
| $l_{sa,0}$ | 2 mm | $\omega_p$ | 250 μm |
| $R_{sa}$ | 5.5 mm | $\delta_0$ | 0.01 |
| $RS_{sa}$ | 5.5 mm | $T_p$ | 260 μs |
| $N_{s0}$ | $3.5 \times 10^{-23}$ m$^{-3}$ | $P_p$ | 15 W |
| $h\nu_p$ | $2.4616 \times 10^{-19}$ J | $t_{qr}$ | 4 ns |
| $\sigma_g$ | $4.3 \times 10^{-22}$ m$^2$ | $t_{qs}$ | 160 ns |
| $\sigma_e$ | $8.2 \times 10^{-23}$ m$^2$ | $R$ | 0.5 |

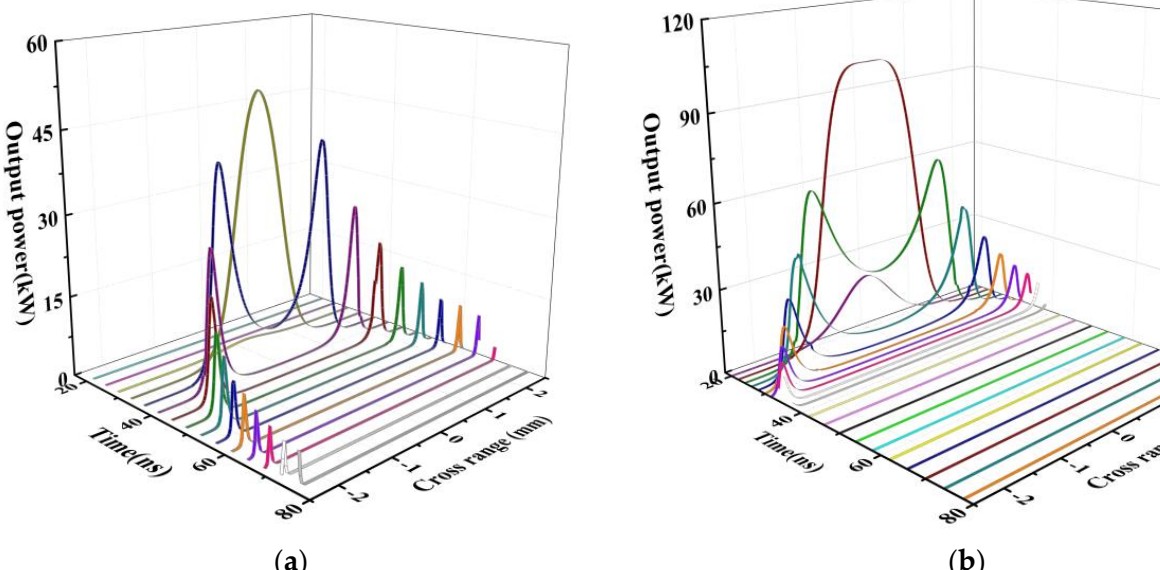

(**a**)          (**b**)

**Figure 4.** Evolution of the radial intensity distribution of the laser output along with the time (**a**) $V_{hl}/V_{\lambda/4} = 1.4$; (**b**) $V_{hl}/V_{\lambda/4} = 1$. (Note: Different colors represent the radial intensity distribution of laser output at different times).

Considering the other three kinds of deformed cylinder saturated absorbers shown in Figure 2, namely Equation (2) with $R_{sa}$ = 3 mm, $RS_{sa}$ = 5.5 mm and $l_{sa,0}$ = 2 mm, Equation (3) with $\beta = 80°$ and $l_{sa,0}$ = 2 mm, as well as Equation (4) with $p = 75$ and $l_{sa,0}$ = 2 mm, the simulations were repeated using these parameters and equations, respectively. Figure 5 shows the predicted radial intensity distributions at the peak of the laser output in the four cases using deformed cylinder saturated absorbers, and another case using normal cylinder saturated absorber. It is apparent that the predicted peak radial intensity distribution of the cylinder saturated absorber with one end cut as a spherical concave surface (Type I SA) is much closer to quasi-super-Gaussian intensity distribution, while the saturated absorbers with one end cut as a spherical convex surface (Type II SA) or a parabolic surface (Type III SA) exhibit weaker spatial homogenization results. Cut as a conical surface (Type IV SA) leads to an opposite effect that further spread the laser intensity to the periphery. To make a more intuitive comparison, Figure 6 shows the radial intensity distribution ratio, defined as the ratio of the central area under the radial intensity distribution curve to the whole area under the same curve, while the central area indicates the corresponding range at 1/e of the highest strength on the cross section. One can find that under the same thickness, Type I SA always leads to a radial intensity ratio around 0.9, which is 0.06 higher than that of Gaussian distribution and is insensitive to the variation of absorber thickness. In addition, the radial intensity distribution ratio of the laser output using Type II SA or Type III SA increases significantly with rising absorber thickness. When the absolute passive

loss inside laser is enhanced enough, the radial intensity distribution ratios of these two cases approach 0.9 also. The radial intensity distribution ratio of the laser using Type IV SA shows opposite behavior. When the absorber thickness is 2.5 mm, a radial intensity distribution ratio of 0.78 is achieved and is 0.06 lower than that of the Gaussian distribution.

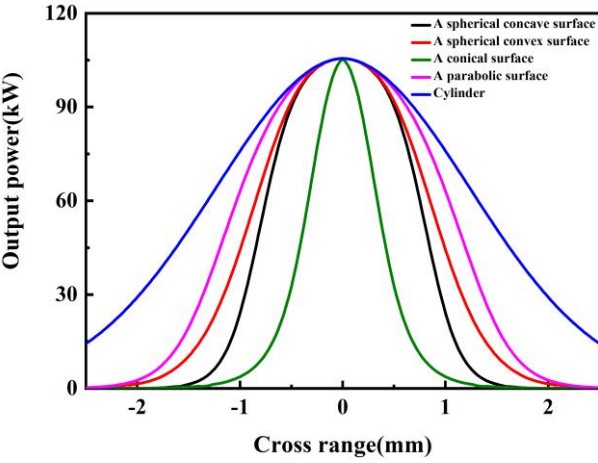

**Figure 5.** Predicted peak radial intensity distribution.

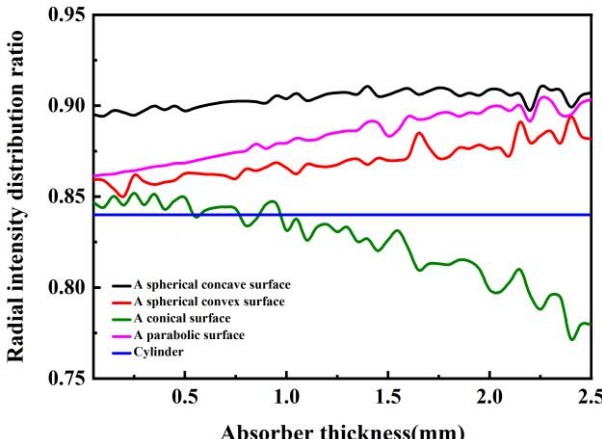

**Figure 6.** Radial intensity distribution ratio of the laser output using 5 different kinds of saturated absorbers.

Pump power is the other parameter possessing global impact on the overall laser behaviors. Figure 7 shows the predicted peak radial intensity distributions ratio of the laser output in the five cases under different pump power when $l_{sa,0}$ is set as 2 mm. For Type I SA, the laser radial intensity distribution ratio still experiences little influence from pump power variation, and the maximum of 0.91 is achieved at a pump power of 15 W. However, the laser radial intensity distribution ratios for Type II SA and Type III SA become worse when the pump power is raised up and quickly decayed to the same results of Gaussian distribution. This phenomenon can be understood when one notices that the saturated absorbers can be saturated more easily at higher laser power. Type IV SA brings out a curious curve that contains several strong fluctuations, this may be due to the fact that the thickness of the saturated absorber changed sharply with radial position in this case, especially in the area around the center of the saturated absorber. In this case, the initial state of the laser intensity distribution can no longer be approximated as Gaussian distribution.

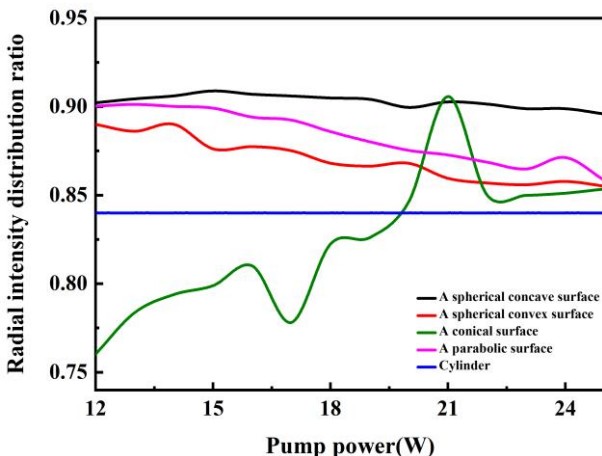

**Figure 7.** Radial intensity distribution ratio of the laser using five different kinds of saturated absorbers versus pump power.

Figure 8 shows the predicted laser peak radial intensity distribution ratio of the laser output in the five cases under different reflectance of the output coupling mirror, when $l_{sa,0}$ is set as 2 mm and $P_p$ is set as 15 W. The rough trends of the five curves are similar to that shown in Figure 7, for the reason that both higher pump power and higher output coupling reflectance will generally cause more intense intracavity laser oscillation. Note that best laser spatial homogenization performance, namely a radial intensity distribution ratio of 0.91, is obtained at a reflectance of 0.55.

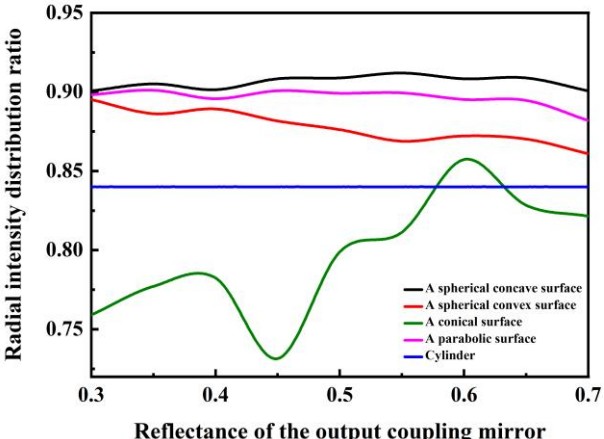

**Figure 8.** Radial intensity distribution ratio of the laser using five different kinds of saturated absorbers versus reflectance of the output coupling mirror.

## 4. Conclusions

In conclusion, a hybrid Q-switching method that relied on deformed cylinder saturated absorbers was proposed, for simultaneous manipulation of the temporal and spatial behaviors of a solid-state pulse laser. The temporal–spatial rate equation model of the laser was given and used to optimize the design parameter of the saturated absorber. The cylinder saturated absorber with one end cut as a concave surface enables the best spatial intensity homogenization. At the case of $R_{sa}$ = 5.5 mm, $RS_{sa}$ = 5.5 mm, $P_p$ = 15 W, $V_{hl}/V_{\lambda/4}$ = 1, the laser PW can be narrowed to be 2.39 ns, and the laser radial intensity distribution was altered to be quasi-super-Gaussian distribution with a radial intensity distribution ratio of 0.91. The nanosecond laser with quasi-uniform intensity distribution and good coherence paid the way for a high efficiency, high energy single mode MOPA. Better manipulation performance of the laser temporal and spatial behaviors can be expected when the parameters of the saturated absorber and resonator design are further optimized.

This kind of laser can be applied in laser amplification, laser material processing, imaging, high efficiency fiber injection, and lithography.

**Author Contributions:** Conceptualization, H.Y. and Y.L.; methodology, K.Z.; software, H.Y. and Y.L.; validation, H.Y. and Y.L.; formal analysis, Y.L. and J.F.; investigation, H.Y. and W.W.; resources, K.Z.; data curation, J.F. and W.W.; writing—original draft preparation, H.Y. and Y.L.; writing—review and editing, Y.L. and K.Z.; visualization, H.Y.; supervision, K.Z.; project administration, K.Z.; funding acquisition, K.Z. All authors have read and agreed to the published version of the manuscript.

**Funding:** This research was funded by National Natural Science Foundation of China (NSFC), grant number 62175135, and Fundamental Research Program of Shanxi Province, grant number 202103021224025.

**Institutional Review Board Statement:** Not applicable.

**Informed Consent Statement:** Not applicable.

**Data Availability Statement:** The data underlying the results presented in this paper are not publicly available at this time but may be obtained from the authors upon reasonable request.

**Conflicts of Interest:** The authors declare no conflict of interest.

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
