# Peer review of "Simultaneous Manipulation of the Temporal and Spatial Behaviors of Nanosecond Laser Based on Hybrid Q-Switching"

_photonics, doi:10.3390/photonics10020227_

Round 1

Reviewer 1 Report

The authors proposed a hybrid Q-switching method based on a special-shaped saturated absorber for simultaneous manipulation of the temporal and spatial behaviors of a solid-state pulsed laser. The temporal-spatial rate equation model of the laser was given. The conditions for realizing the best spatial intensity homogenization performance are specified by simulations. However, the authors should pay some more attention to the rigor of this theoretical work. I give some concerns that may be helpful to improve the manuscript. I recommend the publication of this manuscript once these concerns are addressed in an appropriate way.

1.  Temporal-spatial rate equation model of hybrid Q-switched laser is established by griding the cross-section of the gain crystal and saturable absorber. I am not convinced by this theory. It is well known that the intensity distribution in the cross-section is closely related to the diffraction which is usually described by the wave equations, but I find no information about this concern. This may confuse the readers. For example, for the initial condition, a gaussian beam is supposed, but is it reasonable for all the shape of the saturable absorber? The authors should give some comments to explain how the diffraction is considered.

2. It is better to explain why the manipulation of the spatial intensity distribution is important to let reader understand the motivation of this study. Saying “Nevertheless, the laser spatial intensity distribution manipulation based on the same configuration was seldom investigated” is not enough.

3. Please do some more proof reading to avoid the language problem. As an example, the tense in the sentences “Besides that, there existed a falling edge as long as 45 ns for the photons located on the wings of the cross-section. When an aperture was used to block the photons on the wings, almost 61.5% power loss will be introduced.” is not appropriate.

4.  How did the authors determine the values of ωg and ωsa?

Reviewer 2 Report

The authors present a hybrid Q-switching method based on a special-shaped saturated absorber for which they show simultaneous manipulation of the temporal and spatial behaviors of a solid-state pulse laser. By the use of a temporal-spatial rate equation model of the laser, they optimized the design parameter of the saturated absorber. The paper is well-written and shows interesting results, for this reason, I consider that the document could be accepted after the answer to the next suggestions:

1. I suggest the authors mention the advantages and possible applications of the obtained results.

2. I consider that it is necessary to add a sub-figure 1 about the gain medium and the resonator elements.

3. For equations (5)-(8), I don't see $\delta_{QS}$ in the equations system, maybe it must be $\delta_{EOQS}, could the authors confirm this?

4. I suggest the authors add a table with the values of all the parameters used in the simulations, lines 121-123, to easily use it in the equations system, and this will help with better reproduction of the results. 

5. In lines 171-174, the authors refer that the maximum intensity distribution ratio is 0.91 for pump-power of 15 W for the concave-end-cut cylinder saturated absorber. My question here is about the absorber, because in Figure 1. the black line corresponds to a spherical concave surface, could the authors explain what is the correct surface? 

6. Could the authors mention the advantages of their results versus other commercial or experimental Q-switching systems? Also, could the authors mention possible applications of the implemented method?

Reviewer 3 Report

The authors have reported on a simultaneous manipulation of the temporal and spatial behaviors of a solid-state pulse laser based on a hybrid Q-switching. The temporal-spatial rate equation model was given and used to optimize the design parameter of the saturated absorber in detail. This paper is of high quality in expression. The results of this work will benefit the pulsed laser. I recommend this work be accepted after some minor revisions.

Minor issues:

1. Have the authors considered the influence on laser instability?

2. Whether the temporal-spatial theoretical model used in this paper experiment applies to same similar hybrid Q-switching configurations?

3. Why Nd:YVO4 and Cr:YAG were used in the experiment? Do other saturated absorbers or Nd:YAG crystal being suitable?

Reviewer 4 Report

In the following paper, entitled, “Simultaneous manipulation of the temporal and spatial behaviors of nanosecond laser based on hybrid Q-switching” the Yang et al, reported the hybrid Q-switching method based on a special-shaped saturated absorber for simultaneous manipulation of the temporal and spatial behaviors of a solid-state pulse laser. The results are interesting. I would recommend the manuscript to be accepted after minor revision.

1.      Did the author measured the timing jitter of the laser pulse? Please comment?

2.      The acronym of optical components used in the experimental arrangement needs to be specify in Figure. 1.

3.      Does the role of the length of optical cavity plays a role on the performance and characteristics of the laser pulse?

4.      Any particular reason for using Cr:YAG saturated absorber? Please comment.
